# Exploring the Cognitive Knowledge Structure of Large Language Models: An Educational Diagnostic Assessment Approach

**Zheyuan Zhang**[*], **Jifan Yu**[*], **Juanzi Li**[†], **Lei Hou**
Department of Computer Science and Technology
Tsinghua University, Beijing, 100084, China
{zheyuan-22, yujf21}@mails.tsinghua.edu.cn
{lijuanzi, houlei}@tsinghua.edu.cn

## Abstract

Large Language Models (LLMs) have not only exhibited exceptional performance across various tasks, but also demonstrated sparks of intelligence. Recent studies have focused on assessing their capabilities on human exams and revealed their impressive competence in different domains. However, cognitive research on the overall knowledge structure of LLMs is still lacking. In this paper, based on educational diagnostic assessment method, we conduct an evaluation using MoocRadar, a meticulously annotated human test dataset based on Bloom Taxonomy. We aim to reveal the knowledge structures of LLMs and gain insights of their cognitive capabilities. This research emphasizes the significance of investigating LLMs' knowledge and understanding the disparate cognitive patterns of LLMs. By shedding light on models' knowledge, researchers can advance development and utilization of LLMs in a more informed and effective manner.

## 1 Introduction

Large language models (LLMs), such as GPT series (Brown et al., 2020), Flan (Wei et al., 2022), and PaLM (Chowdhery et al., 2022), have gained significant attention worldwide due to their remarkable ability. Given their unprecedented human-like performances, researchers have started to explore alternative evaluation metrics beyond traditional benchmarks like MMLU (Hendrycks et al., 2021) and Big-Bench (Ghazal et al., 2017).

**Existing Works on LLMs Evaluation with Exams.** Researchers have long sought models capable of passing human exams (Nilsson, 2005). Recently, a new approach simulates professional exams designed for humans to evaluate LLMs. For example, OpenAI (2023) reports the performance of GPT series on a variety of exams, including AP exams, SAT, Leetcode, and so on. There are also emerging

benchmarks that comprise common standardized exams, such as AGIEval (Zhong et al., 2023), C-Eval (Huang et al., 2023), M3Exam (Zhang et al., 2023), and CMExam (Liu et al., 2023). However, although standardized exams contain diverse information, these works condense them into a single overall score, lacking structured understanding of LLMs' knowledge and cognitive patterns.

For example, while LLMs demonstrate exceptional performance on tasks challenging for humans, they might still struggle with basic knowledge, as illustrated in Figure 1, which may lead to over-estimation of the validity of model generated contents. Therefore, there is a pressing need for further research of models' knowledge and cognitive distribution in comparison to humans.

**Proposed Research.** To investigate this problem, we draw inspiration from psychometric methods that use cognitive psychology theories to evaluate LLMs. This topic has gained traction as LLMs continue to demonstrate exceptional performances (Chollet, 2019; Singh et al., 2023; Bubeck et al., 2023). In this work, we adopt the *Educational Diagnostic Assessment* approach and leverage MoocRadar (Yu et al., 2023), a novel student exercise dataset annotated with Bloom's Taxonomy (Anderson and Krathwohl, 2001), to assess the cognitive capability of LLMs. Specifically, we delve into three primary research questions: 1) *Performance Analysis*: the proficiency and robustness of LLMs across various question domains; 2) *Deficit Assessment*: the knowledge structure and the extent to which LLMs are similar with humans; and 3) *Error Assessment*: the error pattern of LLMs in answers and explanations. Our findings contribute to a deeper understanding of the knowledge structure of LLMs and insights for evaluation.

**Contributions.** Our main contributions are:

(i) We introduce the topic of the cognitive knowledge structure of LLMs.

(ii) We propose a method of Educational Diag-

---

[*] Equal contribution
[†] Corresponding Author

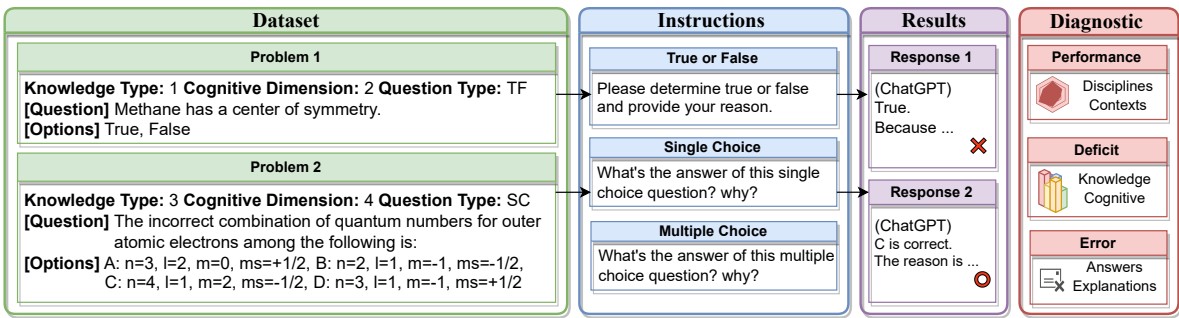

Figure 1: ChatGPT correctly answers a question that is challenging with higher knowledge type and cognitive dimensions right (problem 2) but encounters difficulties in an easier one (problem 1). We design specific instructions to evaluate LLMs and assess their performances in three aspects.

nostic Assessment to evaluate LLMs on their cognitive knowledge structure.

(iii) We assess LLMs' performance, deficits, and errors, gaining insights into their capabilities.

## 2 Method

### 2.1 Educational Diagnostic Assessment

In education scenarios, Diagnostic Assessment is a widely employed method to gauge students' knowledge structure (Falmagne et al., 2006), discovering their proficiency on certain subject matters and learning styles (Vuong et al., 2021), typically through sets of questions (Leighton and Gierl, 2007). Two main approaches of Diagnostic Assessment include **deficit assessment**, which focuses on identifying and addressing knowledge gaps in various domains and the degree of knowledge mastery, and **error assessment**, which focuses on error patterns and strategies for correction (Bejar, 1984). Drawing inspiration from Diagnostic Assessment methods, in this work, based on Bloom's Taxonomy, we use deficit assessment to test the accuracy of models on a wide range of exercises, and error assessment on their answers and explanations.

### 2.2 Experimental Setup

**Dataset.** In this section, we introduce the dataset utilized for assessment, MoocRadar, offering an extensive overview of its general information. MoocRadar is a fine-grained and multi-aspect exercise repository designed for cognitive modeling and educational diagnostic assessment. According to Bloom's Taxonomy, questions in MoocRadar are categorized into four Knowledge-Types: Factual-knowledge, Conceptual-knowledge, Procedural-knowledge, and Meta-knowledge; and six cognitive dimensions: Remember, Understand, Apply, Ana-

lyze, Evaluate, and Create. Table 1 demonstrates a detailed description of Bloom's Taxonomy.

| Cognitive Dimensions | Descriptions |
|---|---|
| Remembering | Remember, Know, Identify, ... |
| Understanding | Translate, Explain, Induce, ... |
| Applying | Prove, Estimate, Execute, ... |
| Analyzing | Compare, Select, Organize, ... |
| Evaluating | Evaluate, Judge, Criticise, ... |
| Creating | Design, Create, Program, ... |
| **Knowledge Types** | **Descriptions** |
| Factual | Terminology and Details |
| Conceptual | Relationships and Theories |
| Procedural | Processes and Methods |
| Meta | Strategy and Self-knowledge |

Table 1: A detailed descriptions and examples of Bloom's Taxonomy in MoocRadar.

We carefully select 8453 questions appropriate for model evaluation, which fall into three types: single choice (SC), multiple choice (MC), and true or false (TF). Additionally, we exclude the dimension of Create because of the scarcity of related exercises. We further classify these questions into four disciplines by their course information, including STEM, social science, humanity, and others. We test the performance of models on them and analyze the distribution of these features. More details of MoocRadar are illustrated in the appendix.

**Model Selection.** We carefully choose 3 advanced models that have consistently demonstrated leading performance and are widely recognized in the field, including: *Text-Davinci-003*, *Chat-GPT*, and *GPT-4*, which represent a series of most acknowledged models. All experiments are performed using the APIs provided by OpenAI. Specif-

ically, we use the completion API for Text-Davinci-003 and the chat completion API for ChatGPT and GPT-4. To ensure consistency in the quality of the responses, we set the temperature to 0 to get greedy search responses generated by each model.

**Experimental Design.** As shown in Figure 1, we design different prompts tailored to each type of exercises to query LLMs for both answers and explanation. All tasks are conducted in zero-shot scenario. To simulate human-like behavior that solving exercises with relevant knowledge, we leverage the BM25 algorithm to retrieve the two most related discussions from the subtitles in the corresponding courses in MOOCCubeX (Yu et al., 2021) and test their effect. Moreover, we extract real student behaviors on MoocRadar dataset from MOOCCubeX and calculate their average scores to serve as a reference of humans. Based on both results from human and LLMs, this work provides a road map with investigation to the following research questions:

**(RQ1) Performance Analysis:** What's the features of LLMs' basic performance on different disciplines and their robustness to these questions?

**(RQ2) Deficit Assessment:** According to Bloom Taxonomy, compared with humans, what knowledge distribution does LLMs demonstrate? Are they similar to humans in knowledge structure?

**(RQ3) Error Assessment:** Based on answers and explanations, what's their pattern of errors?

## 3 Experiment

In this section, we conduct experiments and analyze the results from three perspectives in the following subsections. We assign a score of 1 to each question type. Following standardized exams, for multiple questions, models receive a score of 0.5 if they fail to select all correct options. We then calculate the average score across questions.

### 3.1 Performance Analysis

Firstly, we assess their performance both with and without contexts, compare their performance in different disciplines, and examine their robustness.

**Disciplines and Context.** We exhibit scores of model answers with or without context on the four disciplines (STEM, social science, humanity, and others). As shown in Table 2, the later versions of GPT significantly outperform previous models, with GPT-4 being the most advanced, but not better than humans' average. Additional knowledge from context indeed enhances the performance of

the models. Comparatively, STEM exercises are more challenging as illustrated in human results, while LLMs demonstrate impressive capability in STEM knowledge. GPT-4 even outperforms humans with context. However, it is surprising that LLMs don't perform as effectively in social science and humanities exercises, even though these disciplines primarily involve natural language.

| Models | Total | STEM | S.S. | Human. | Others |
|---|---|---|---|---|---|
| GPT-3.5 | 0.436 | 0.418 | 0.461 | 0.483 | 0.421 |
| -context | 0.508 | 0.468 | 0.554 | 0.614 | 0.504 |
| ChatGPT | 0.506 | 0.480 | 0.547 | 0.533 | 0.525 |
| -context | 0.526 | 0.441 | 0.642 | 0.639 | 0.601 |
| GPT-4 | 0.657 | 0.613 | 0.732 | 0.684 | 0.690 |
| -context | 0.687 | **0.629** | 0.765 | 0.774 | 0.733 |
| Human | **0.746** | 0.625 | **0.924** | **0.854** | **0.791** |

Table 2: Models and human performance. S.S. and Human. are short for social science and humanity, and context is the result of models with context. Bold figures denote best performance.

**Robustness.** In single choice questions, we manipulate the order of the options by either placing the correct answer at the beginning or the end. This allows us to examine if such modifications affect the model's accuracy. As shown in table 3, we find that 1) ChatGPT is more robust to changing of options, while the other two exhibits a cognitive bias as *Primacy Effect* (Deese and Kaufman, 1957) that early appearance aids performance; 2) if the correct answer appears later in GPT-3.5 and GPT-4, they mistakenly change their answers and explanations; 3) later appearance causes less consistency in answers and explanations in less robust models.

| Models | Answer | Explanation | Real |
|---|---|---|---|
| GPT-3.5 | 0.405 | 0.436 | 0.390 |
| -first | **0.576** | **0.554** | **0.476** |
| -last | 0.362 | 0.390 | 0.296 |
| ChatGPT | **0.489** | **0.501** | **0.487** |
| -first | 0.440 | 0.438 | 0.438 |
| -last | 0.412 | 0.430 | 0.412 |
| GPT-4 | 0.635 | 0.643 | 0.632 |
| -first | **0.727** | **0.728** | **0.727** |
| -last | 0.520 | 0.529 | 0.473 |

Table 3: Single choice accuracy of three models. First and last indicate the place of the correct answers. Real is when answers and explanations are both correct. Bold figures denote the best performance.

## 3.2 Deficit Assessment

We utilize Bloom's Taxonomy in MoocRadar to demonstrate models' distribution in cognitive dimensions and knowledge types and design a score to measure similarities of models to humans.

**Bloom Taxonomy Distribution.** As shown in Figure 2, we demonstrate the distribution based on Bloom's taxonomy, where deeper colors represent better performance. The 0 and 1 grids are due to the limited number of exercises, typically only one. Generally, both in knowledge types and cognitive dimensions, questions in the intermediate range are more challenging for models and humans. We design a similarity score for deeper understanding.

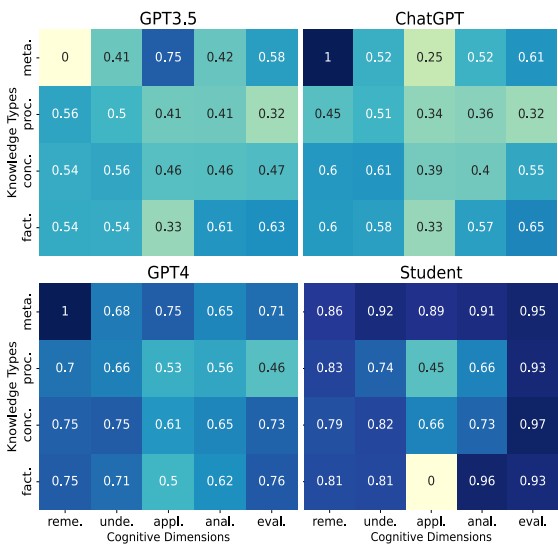

Figure 2: The distributions of accuracy in Bloom's Taxonomy of different models and average of students.

**Similarity Score.** According to the accuracy of models in various dimensions of knowledge and cognition, we develop a metric to measure their similarity to humans, which primarily considers knowledge structure, beyond mere performance, and estimates the extent to which their cognitive structure is proportional to that of humans. Specifically, given a model $M$, the 4*5 vector of the model distribution in bloom's taxonomy $x$ and human distribution $y$, convert $x$ and $y$ into 1*20 vectors $\tilde{x}$ and $\tilde{y}$, the similarity between $M$ and human can be defined as: $Likeness(M) = \rho(\tilde{x}, \tilde{y})$, where $\rho(\tilde{x}, \tilde{y})$ represents the Pearson Correlation Coefficient of $\tilde{x}$ and $\tilde{y}$. We calculate the $Likeness$ of the three models in Table 4. The likeness also exhibits a rising tendency as the models evolve. Models that follow human instructions better are also more similar to humans in knowledge structure.

| Models | GPT-3.5 | ChatGPT | GPT-4 |
|---|---|---|---|
| Likeness | 0.262 | 0.396 | 0.474 |

Table 4: Models' similarity to human, measured by Pearson Correlation Coefficient of knowledge structures.

## 3.3 Error Assessment

In this section, we analyze the error of each models, by delving into their explanation of their answers.

**Explanation Accuracy.** Table 5 demonstrate the accuracy of answers in each type. We mainly find that: 1) Models perform best on TF and worst on MC. MC could be more difficult than SC and TF, because of more thinking steps (determine TF of each options, and select multiple ones). 2) Explanations and answers are more consistent in TF than in SC and MC for the same reason, as there are more chances to make errors. 3) Accuracy of explanations falls behind answers in MC, where models can select some of the correct options for the wrong reason. 4) Context does not necessary aid and even hurt explanation performances, but indeed aids answer accuracy. More advanced models are more consistent in their answers and explanations.

| Models | TF | SC | MC |
|---|---|---|---|
| GPT-3.5 | 0.531 (0.531) | 0.437 (0.405) | 0.383 (0.483) |
| -context | 0.617 (0.617) | 0.376 (0.480) | 0.280 (0.527) |
| ChatGPT | 0.617 (0.592) | 0.501 (0.488) | 0.298 (0.496) |
| -context | 0.715 (0.706) | 0.492 (0.482) | 0.333 (0.541) |
| GPT-4 | 0.751 (0.746) | 0.643 (0.635) | 0.555 (0.663) |
| -context | 0.775 (0.771) | 0.605 (0.667) | 0.578 (0.691) |

Table 5: Accuracy of explanations. TF, SC, and MC are short for the three question types. The numbers in parentheses represent answer accuracy.

## 3.4 Discussion

In this section, we discuss our findings on the proposed three research questions:

**Performance Analysis.** We exhibit different models' performance. Comparing with humans, they are less proficient in disciplines primarily involve natural language, but better at STEM. Though with sufficient knowledge, they might have hallucination on specific long-tail concepts in humanity and social science. LLMs are not robust in option orders, and exhibit a cognitive bias as *Primacy Effect* rather than Recency Effect.

**Deficit Assessment.** Models are less proficiency in the intermediate range of Bloom's Taxonomy. The reason could be that application-based ques-

tions, such as solving mathematical problems and making deductions using chemical theorems, are prone to errors and are inherently challenging for models. For analyzing and evaluating questions, the strong linguistic capabilities of models allow them to excel in these tasks, and perform even better than intermediate-level questions. More advanced models demonstrate more similarity with humans in knowledge structure, which might be an additional effect of human alignment.

**Error Assessment.** By comparing different kinds of questions, we find that gap exists for models between knowledge and answers. They perform worse in multiple choices, as there are more thinking steps and error chances. Accuracy of explanations can be worse than answers: as models were asked to generate answers first, their explanation could shift due to wrong answers and question orders, and cause their hallucinations. Due to the limitations of autoregressive architecture (Bubeck et al., 2023), their errors could snowball.

## 4 Conclusion

In this work, we introduce a new research question on LLMs analyzing, which calls for a deeper understanding of the knowledge structure of these models. We use Educational Diagnostic Assessment as a tool to test the performance of LLMs on various dimensions, and develop a metric to measure the similarity of their knowledge structure with humans. We provide findings and discussion for insight into research on the cognition of LLMs.

## Limitations

In this section, we describe the limitations of this work in terms of the dataset and experiments.

**Dataset.** We investigated the knowledge distribution of LLMs based on the MoocRadar dataset. MoocRadar is a fine-grained, well-structured dataset that is distinct from commonly used benchmarks in terms of knowledge annotation. However, as a dataset for educational diagnostic assessment, it's still limited in the following aspects: 1) Different categories of exercises (e.g. question type, disciplines) have an unbalanced distribution; 2) As demonstrated in the Robustness section, the performance of the models can vary due to different forms of exercises.

**Experiment.** Due to time and cost constrains, 1) we only included three LLMs by OpenAI, which are all closed-source models. Therefore, we did not conduct experiments at the parameter level. 2) though we have discovered some phenomena, further experiments and deeper analysis are not conducted. We include some of them in the case study section in the appendix.

**Future Works.** Future works include 1) more models for experiments, 2) further exploration on robustness and similarity with humans, and 3) as the next step of diagnostic assessment, investigate how to optimize the knowledge structure of LLMs.

## Ethics Statement

We foresee no ethic concerns in this work. The MoocRadar dataset employed in our research is publicly available, and it does not contain any personal information.

## Acknowledgements

This work is supported by a grant from the Institute for Guo Qiang, Tsinghua University (2019GQB0003), and also supported by Tsinghua University Initiative Scientific Research Program.

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

# A   Appendix

## A.1   Details of Experiment

This subsection shows details of the dataset we use, and experiment for diagnostic assessment.

**Problem Statistics.** Table 6 demonstrates the details of the dataset we use, which is selected from the original MoocRadar. Generally, we include three question types (single choice, multiple choice, and true or false), four knowledge types (factual knowledge, conceptual knowledge, procedural knowledge, and meta knowledge), and five cognitive dimensions (remember, understand, apply, analyze, and evaluate), to form a total dataset of 8430 questions.

| Ex. | $N$ | Knowledge Types | $N$ | Cognitive Dimensions | $N$ |
|---|---|---|---|---|---|
| SC | 5968 | Factual | 2020 | Remember | 1715 |
| MC | 1086 | Conceptual | 4032 | Understand | 4066 |
| TF | 1376 | Procedural | 2268 | Apply | 1667 |
| total | 8430 | Meta | 110 | Analyze | 640 |
| / | / | / | | / | Evaluate | 342 |

Table 6: Categories of data. Ex., SC, MC, and TF are short for Exercise, Single Choice, Multiple Choice and True or False questions.

| Types | Questions | Options | Answers |
|---|---|---|---|
| SC | Which of the following works was created by Vincent van Gogh? | A: Les Nymphéas 
 B: Sunflowers 
 C: Grande Odalisque | B |
| MC | Which of the following are rare earth elements? | A: Uranium 
 B: Lutecium 
 C: Dysprosium 
 D: Samarium | B, C, D |
| TF | Light only exhibits the properties of waves. | True 
 False | False |

Table 7: Question types examples: single choice (SC), multiple choice (MC), and true or false (TF). SC have only one correct option, while MC have 2 or more than 2 correct options. TF should be determined as True or False.

**Problem Examples.** Table 7 demonstrates examples for each type of questions. There are two or more than two options in single choices and only one correct options, while multiple choices have more than one correct options. True or false questions should be answered as True or False.

**Querying Details.** For context settings, we use the BM25 algorithm to retrieve the two most related contexts from the subtitles of the corresponding class. As illustrated in Figure 3, for the question about the pioneer of mathematical logic, the BM25 algorithm retrieves context about the emergence and development of logic and the concept of mathematical logic. The two contexts will be placed before instruction, along with the questions and options to form the prompt, and fed into LLMs. In non-context settings, the context position will simply be empty. We also test different instructions to make sure that models will follow them to provide answers and explanations.

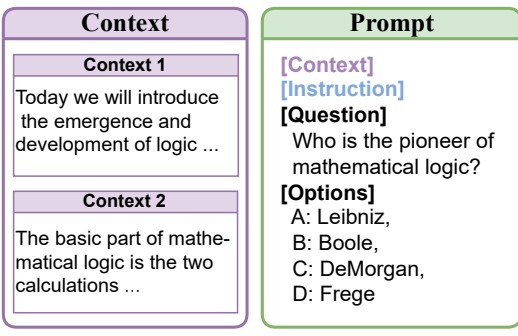

Figure 3: Construction of prompts in experiments.

**Annotation details.** To accurately assess the models' performance on both answers and explanations, we first invited educational experts to filter questions, because a part of questions in the dataset have accompanying pictures, which can't be input into models. Then we asked human annotators to evaluate the answers and explanations generated by the three models respectively, to prevent misjudgment in automatic evaluation tools. These annotators are familiar with MOOC exercises, and provided with the original questions and correct answers, and have access to the internet to determine accurately.

## A.2   Cases

In this subsection, we demonstrate some of the error cases we have seen during our experiments, and hope to provide insights into the model's error patterns.

| | |
|---|---|
| **Question** | If all birds can fly, then **ostriches** can fly. **Ostriches** are birds. Therefore, **ostriches** can fly. Does this reasoning contain any **formal errors**? |
| **Correct Answer** | No. |
| **Model Answer (ChatGPT)** | **Yes**. ... While the premises state that every bird is capable of flying, it does not mean that every individual bird, including the ostrich, is capable of flying. **Ostriches are actually flightless birds**. |
| **Question** | If all birds can fly, then **sparrows** can fly. **Sparrows** are birds. Therefore, **sparrows** can fly. Does this reasoning contain any formal errors? |
| **Correct Answer** | No. |
| **Model Answer (ChatGPT)** | **No**. The reasoning is an example of a valid deductive argument ... **from a purely logical standpoint**, the argument itself is valid. |

Table 8: **Case 1: Changing the entities infects the answer.** If the question is consistent with its knowledge, ChatGPT indeed follows the task of focusing on the formal errors or purely logic. Otherwise, it fails to follow the question, as it refuses to override its own knowledge. This phenomenon indicates ChatGPT might be less robust when there are knowledge conflicts.

| | |
|---|---|
| **Question** | "When undergoing uniform acceleration, the direction of velocity always lies along the same line as the direction of acceleration." Is it correct? **A. No**, **B. Yes** |
| **Correct Answer** | **A. No** |
| **Model Answer (GPT-4)** | The answer is **A: No**. The reason is that uniform acceleration only requires the acceleration to remain constant, but the direction of velocity and acceleration can be different. For example, in uniform circular motion, ... |
| **Question** | "When undergoing uniform acceleration, the direction of velocity always lies along the same line as the direction of acceleration." Is it correct? **A. Yes**, **B. No** |
| **Correct Answer** | **B. No** |
| **Model Answer (GPT-4)** | The answer is **A. Yes**. This is because uniform acceleration refers to ... In this case, the change in velocity is aligned with the direction of acceleration. |

Table 9: **Case 2: the *Primacy Effect* of GPT-4.** For the exactly same questions, switching options causes a direct change in GPT-4's answer, and both answers are accompanied with solid but conflict explanations, which proves the instability of its knowledge. Statistically, models have a tendency to choose more forward contents, which influence their rationale more, just like the Primacy Effect in cognitive psychology. This phenomenon is also observed in GPT-3.5.

| | |
|---|---|
| **Question** | The term "logic" originally derives from: A: English, B: French, **C: Latin**, **D: Greek**. |
| **Correct Answer** | **D: Greek** |
| **Model Answer (ChatGPT)** | The term "logic" originally derives from the language: **C.Latin**. The term "logic" originated from the Latin word "logica" which itself **was derived from the Greek word "logikē."** Therefore, ... |
| **Question** | The term "logic" originally derives from: A: English, B: French, **C: Greek**, **D: Latin**. |
| **Correct Answer** | **C: Greek** |
| **Model Answer (ChatGPT)** | The term "logic" originally derives from: **C: Greek**. The Greek philosopher Aristotle is often credited as one of the founders of formal logic and his work on reasoning and ... |
| **Question** | Which language does the term 'logic' originally derives from? |
| **Correct Answer** | **Greek** |
| **Model Answer (ChatGPT)** | The term "logic" originally derives from the **Greek** word "logikē," which is derived from "logos," meaning "word," "reason," or "thought." The study of logic can be traced back to ... |

Table 10: **Case 3: Easier question forms aid performances.** We have discussed the effect of different question types. ChatGPT have the correct knowledge about the origin of this word, but it cannot give correct answers, as the first question demonstrates. The following two questions types improve the performance of ChatGPT, by two different ways: moving the correct option forward which is consistent to Case 2, or ease the burden of models to answer by simplifying the question form. This case corroborates the findings that models are better at TF than SC or MC, because there are fewer thinking steps.