# OpenReview forum: "Exploring the Cognitive Knowledge Structure of Large Language Models: An Educational Diagnostic Assessment Approach"
_EMNLP/2023/Conference — EMNLP 2023 Findings_

### Official Review · Reviewer_rWPM · 2023-08-04

**Soundness:** 3

**Excitement:**

4: Strong: This paper deepens the understanding of some phenomenon or lowers the barriers to an existing research direction.

**Paper Topic And Main Contributions:**

This paper proposed an educational diagnostic assessment approach to test the performance of LLMs on various dimensions. Current works lack a structured understanding of LLMs’ knowledge and cognitive patterns. This approach conducts an evaluation using MoocRadar, a meticulously annotated human test dataset based on Bloom Taxonomy, and delves into three primary research questions: Performance Analysis, Deficit Assessment and Error Assessment.
There are three main contributions: this work introduces the topic of the cognitive knowledge structure of LLMs; it proposes a method of Educational Diagnostic Assessment to evaluate LLMs on their cognitive knowledge structure; it assesses LLMs’ performance, deficits, and errors, gaining insights into their capabilities.


**Reasons To Accept:**

Strength 1: This paper is clearly organized and well-written. It does not create barriers for the readers and is very clear and logical in its presentation.

Strength 2: The motivation is clear and the research question is valuable. This paper explores LLMs' knowledge structures and cognitive patterns and compares them with human cognition. This will help the progress of LLMs.

Strength 3: The experimental methodology is presented very clearly, from three perspectives: Performance Analysis, Deficit Assessment and Error Assessment. The content is very detailed and the conclusion of the paper is relatively reasonable.

Strength 4: The results of the experiment are very complete and illustrate the strengths and weaknesses of the current LLMs in answering questions and making explanations from different perspectives.


**Reasons To Reject:**

Weakness 1: In fact, different instructions have a significant effect on the results of LLMs. The authors need to add an analysis of the effect of the selected instructions on the current experimental results.

Weakness 2: Such an approach cannot be effectively extended to other datasets, which leads to limitations in exploring the structure of knowledge.

Weakness 3: I suggest that the author can describe the MoocRadar dataset more carefully (e.g., Knowledge-Types). This may be a barrier for readers.


**Reproducibility:**

3: Could reproduce the results with some difficulty. The settings of parameters are underspecified or subjectively determined; the training/evaluation data are not widely available.

**Reviewer Confidence:**

4: Quite sure. I tried to check the important points carefully. It's unlikely, though conceivable, that I missed something that should affect my ratings.

---

> ### Author Rebuttal · Authors · 2023-08-28
>
> Thank you for your review and constructive suggestions. We also genuinely appreciate your recognition of our experimental methodology, presentation, and readability. Here we would like to take this opportunity to address the concerns you've raised.
>
> > Weakness 1: In fact, different instructions have a significant effect on the results of LLMs. The authors need to add an analysis of the effect of the selected instructions on the current experimental results.
>
> Thank you for pointing that out, and we believe this is a crucial question. This question encompasses two subsidiary questions: 1) Can our instructions be accurately comprehended by models? 2) How could our instructions affect the current experimental results?
>
> * Regarding the first question, before conducting the formal experiments, we assessed three candidate prospective prompts, by examining whether models can follow the instructions to generate answers and explanations. For example, in single-choice questions, both the choice and the explanation should be generated. We sampled 100 questions to examine three candidate prompts, and the results are shown below:
>
> |      | GPT3.5 | ChatGPT | GPT4  |
> |------|--------|---------|-------|
> | Prompt1 (Ours)                                    |
> | What's the answer to this single-choice question? Why? | 1.00 (0.99) | 1.00 (1.00) | 1.00 (1.00) |
> | What's the answer to this multiple-choice question? Why? | 1.00 (1.00) | 1.00 (1.00) | 1.00 (1.00) |
> | Please determine true or false and provide your reason. | 1.00 (1.00) | 1.00 (1.00) | 1.00 (1.00) |
> | Prompt2                                           |
> | Which of the following options in the given questions is the correct answer? The reason is?  | 1.00 (0.28) | 1.00 (1.00) | 1.00 (1.00) |
> | Which of the following options in the given questions are the correct answers? The reason is? | 1.00 (0.44) | 1.00 (1.00) | 1.00 (1.00) |
> | Is the statement of the following questions true or false? The reason is? | 1.00 (1.00) | 1.00 (1.00) | 1.00 (1.00) |
> | Prompt3                                           |
> | Please tell me the answer to this single-choice question and provide your reason. | 1.00 (0.99) | 1.00 (1.00) | 1.00 (1.00) |
> | Please tell me the answer to this multiple-choice question and provide your reason. | 1.00 (1.00) | 1.00 (1.00) | 1.00 (1.00) |
> | Please tell me the answer to this true or false question and provide your reason. | 1.00 (1.00) | 1.00 (1.00) | 1.00 (1.00) |
>
>
> The numbers and the numbers within parentheses represent the probabilities assigned by the model to the provided answers and explanations, respectively. With our instructions, answers are correctly derived from the choices provided and reasons were given, which substantiates that models accurately comprehend them.
>
> * As for the second question, we also find this to be both interesting and significant. For example, in our selected instructions, we ask models to output answers before they give explanations. We have provided some insights in the Discussion section, where we conjecture models' explanations could shift due to wrong answers and question orders, as they have to give answers before rationales. This leads to a lower accuracy of explanations. Thank you for reminding us, and we intend to incorporate an extended analysis in the additional page if this paper could be accepted.
>
> > Weakness 2: Such an approach cannot be effectively extended to other datasets, which leads to limitations in exploring the structure of knowledge.
>
> Thank you for your insightful question. In this work, our rationale for using the MoocRadar dataset stems from its detailed question annotations. It is its granularity of cognitive dimension annotating that has enabled us to achieve a deeper understanding of the cognitive capabilities of LLMs. As a short paper, we aimed to present insights, with the hope that in the future, more datasets with similarly detailed annotations on knowledge structure will emerge. This advancement will undoubtedly further propel the development of this field.
>
> > Weakness 3: I suggest that the author can describe the MoocRadar dataset more carefully (e.g., Knowledge-Types). This may be a barrier for readers.
>
> As a brief supplement, the categorization according to the Bloom's Taxonomy is as follows:
>
> **Cognitive Dimensions** are: *Remembering* (Remember, Know, Identify), *Understanding* (Translate, Explain), *Applying* (Prove, Execute), *Analyzing* (Select, Compare, Summarize), *Evaluating* (Evaluate, Judge, Criticise), and *Creating* (Create, Design).
>
> **Knowledge Types** are: *Factual Knowledge* (knowledge of terminology and details), *Conceptual Knowledge* (knowledge of relationships and theories), *Procedural Knowledge* (knowledge of processes and methods), and *Metacognitive Knowledge* (knowledge of strategy, self-knowledge and "thinking about thinking"). [1, 2]
>
> Thank you again for your suggestion. If this article is accepted, we will provide more comprehensive details about the dataset in the additional page to further improve our readability.
>
> **[Final Remark]** We would like to reiterate our gratitude. Thank you for your insightful recommendations that will contribute to the refinement of our work. We hope our supplementary information can address your concerns. If feasible, we would greatly appreciate your consideration in enhacing the paper's soundness score. If you have more questions, we are more than willing to engage in further discussions.
>
> Ref:
>
> [1] Yu, Jifan, et al. "MoocRadar: A Fine-grained and Multi-aspect Knowledge Repository for Improving Cognitive Student Modeling in MOOCs." in Proceedings of SIGIR, 2023.
>
> [2] Anderson, Lorin W., and David R. Krathwohl. A taxonomy for learning, teaching, and assessing: A revision of Bloom's taxonomy of educational objectives. Longman, 2021.

---

### Official Review · Reviewer_Hzpw · 2023-08-04

**Paper Topic And Main Contributions:** 1. Based on educational diagnostic as…
**Soundness:** 3

**Excitement:**

4: Strong: This paper deepens the understanding of some phenomenon or lowers the barriers to an existing research direction.

**Reasons To Accept:**

1. This is an interesting study that reveals the knowledge structure of LLMs.

2. This paper designs specific instructions to evaluate LLMs and assess their performances in three aspects.

3. The authors carries out experimental design from multiple perspectives to evaluate LLMs more comprehensively and effectively.


**Reasons To Reject:**

1. The description of the method is too short, the description of diagnostic assessment should be added.

2. The work needs to be evaluated against more models, resulting in more comprehensive analysis and discovery.

**Reproducibility:**

3: Could reproduce the results with some difficulty. The settings of parameters are underspecified or subjectively determined; the training/evaluation data are not widely available.

**Reviewer Confidence:**

3: Pretty sure, but there's a chance I missed something. Although I have a good feel for this area in general, I did not carefully check the paper's details, e.g., the math, experimental design, or novelty.

---

> ### Author Rebuttal · Authors · 2023-08-28
>
> Thank you for acknowledging the novelty of our work and for providing constructive feedback on our method and experiments. We are deeply honored by the opportunity to enhance our work with your assistance and address some of your concerns.
>
> > The description of the method is too short, the description of diagnostic assessment should be added.
>
> We appreciate your suggestion to fill in more details. Educational Diagnostic Assessment is a widely adopted method used in education to evaluate a student's knowledge and cognitive level, such as strengths, weaknesses, and learning styles [1]. In pursuit of a deeper comprehension of the knowledge and cognitive structure of LLMs, we drew inspiration from Educational Diagnostic Assessment to organize our experiments. To be precise, we adopt two traditional methods in Diagnostic Assessment, Deficit Assessment and Error Assessment [2], respectively focusing on the degree of knowledge mastery and the error patterns by examinations. We transpose them into the scenario of LLMs, devising experiments that respectively examine the knowledge structure and error types with explanations. This allows for a comprehensive assessment of the model's performance.
>
> Should this work be accepted, we intend to include more refined conceptual definitions on the additional page. We once again express our gratitude for your valuable suggestions.
>
> > The work needs to be evaluated against more models, resulting in more comprehensive analysis and discovery.
>
> We perceive this as an excellent suggestion, especially in light of the current proliferation of powerful and diverse LLMs. We are indeed inclined to delve deeper into this realm. As a short paper, constrained by both length and resources, we employed three models that are currently at the forefront of capability and exhibit continuity in development. Our primary aspiration is to present a sharp insight that can attract more exceptional endeavors in this domain.
>
> **[Final Remark]** Thank you again for assisting us to improve our work. We hope our response has addressed your concerns about methods and models. We would really appreciate it if you could consider raising the score. If you have more questions, please feel invited to engage with us for further discussions.
>
>
> Ref:
>
> [1] Falmagne, J-C., et al. "The assessment of knowledge, in theory and in practice." IEMC'03 Proceedings. Managing Technologically Driven Organizations: The Human Side of Innovation and Change. IEEE, 2003.
>
> [2] Bejar, Isaac I. "Educational diagnostic assessment." Journal of educational measurement, 1984.

---

### Official Review · Reviewer_XMhD · 2023-08-04

**Soundness:** 4

**Excitement:**

4: Strong: This paper deepens the understanding of some phenomenon or lowers the barriers to an existing research direction.

**Paper Topic And Main Contributions:**

This paper evaluated the OpenAI LLMs' performance in an educational assessment context, and how the models' outputs align with the cognitive psychology framework, like Bloom's Taxonomy. The authors showed that the models are better than humans in STEM but not in other topics, less proficient in questions located in the intermediate range of Bloom's Taxonomy, and performing less in multiple-choice questions than T/F or single-choice questions.

**Questions For The Authors:**

A. In Section 2.2 (Lines 140-145), the authors specified that "all tasks are conducted in zero-shot scenario." However, in the same section, the authors also said, "leverage the BM25 algorithm to retrieve the two most related discussions from the subtitles in the corresponding courses". The authors in Appendix A (Lines 455-468) iterate that they tested in-context and non-context settings. These sounds contradict the first explanation of using "zero-shot scenario." Can you clarify this?

B. In Figure 2, do numbers represent students' performance/accuracy in corresponding questions?

C. Can you add an extended version of Table 4 that includes ±context, ordering effect, and topic subjects? I think it will be very interesting.

D. Can you add your insights in the Discussion section on why the LLMs perform worse in intermediate-level Bloom's Taxonomy questions and how to improve the models' performance?

**Reasons To Accept:**

- More details about the annotations for the dataset and models' answer/explanation accuracies would be important to ensure the credibility of the results. For example, how many annotators rated how many questions? What were the inter-rater scores? etc.
- How the models generated explanations? Can you share the prompts?

**Reasons To Reject:**

Although the results are interesting, and their usage of Bloom's taxonomy was really insightful, I think the paper missed essential details about the experiment settings and reproducibility.
- More details about the annotations for the dataset and models' answer/explanation accuracies would be important to ensure the credibility of the results. For example, how many annotators rated how many questions? What were the inter-rater scores? etc.
- How the models generated explanations? Can you share the prompts?

**Reproducibility:**

3: Could reproduce the results with some difficulty. The settings of parameters are underspecified or subjectively determined; the training/evaluation data are not widely available.

**Reviewer Confidence:**

3: Pretty sure, but there's a chance I missed something. Although I have a good feel for this area in general, I did not carefully check the paper's details, e.g., the math, experimental design, or novelty.

**Typos Grammar Style And Presentation Improvements:**

- Adding horizontal lines in Table 4 will improve the readability and consistency with other tables.

---

> ### Author Rebuttal · Authors · 2023-08-28
>
> We greatly appreciate the comprehensive and valuable suggestions and questions you've raised, as well as your recognition of the excitement of our work. We kindly respond to your concerns as follows.
>
> > Details about experiments: More details about the annotations for the dataset and models' answer/explanation accuracies would be important to ensure the credibility of the results. For example, how many annotators rated how many questions? What were the inter-rater scores? etc.
>
> Thank you for pointing out this important issue. In particular, we initially engaged a group of experts to conduct two rounds of screening and annotations on each question in the dataset. The aim was to eliminate questions containing images to answer (which can't be models' input), unanswerable questions, and questions without answers (e.g., vote questions), reducing the total number of questions from 9328 to 8430. Subsequently, we enlisted a team familiar with the MOOC context. Each team member was responsible for either 600 or 1000 questions, contingent on the question type. We presented the standard answers and contexts for each question to the team members and encouraged them to verify the model's outputs for correctness using online resources. After one round of annotation, team members exchanged and performed mutual checks and corrections on each other's work.
>
> We have exhibited part of the annotation details in the Appendix. Once the paper is accepted, we also intend to supplement additional content on the additional page to improve the paper further.
>
> > How the model generated explanations? Can you share the prompts?
>
> Thank you for your reminder. We exhibit our instructions in Figure 1. For example, for True or False questions, we prompt the model with "Please determine true or false of the following question and provide your reason." We further examine the effectiveness of our instruction, to ensure that models indeed provide their answers and explanations based on the prompts. We will also consider providing more detailed examples in the public code repository and appendix.
>
> > Question A: In Section 2.2 (Lines 140-145), the authors specified that "all tasks are conducted in zero-shot scenario." However, in the same section, the authors also said, "leverage the BM25 algorithm to retrieve the two most related discussions from the subtitles in the corresponding courses". The authors in Appendix A (Lines 455-468) iterate that they tested in-context and non-context settings. These sounds contradict the first explanation of using "zero-shot scenario." Can you clarify this?
>
> This is a great question, and we will provide further clarification here. Regarding the term "zero-shot", we are referring to the fact that we did not incorporate an in-context learning method [1] when generating model outputs. Instead, we directly provided the model with the question without pre-exposing it to examples of single-choice or multiple-choice answers. On the other hand, within our contextual setup, we indeed introduced additional question-related knowledge to the model. As depicted in Figure 3 of the appendix, we augmented the question stem by including subtitles from the course, to simulate students' behaviors.
>
> Therefore, while both our context and in-context learning (ICL) employ the term "context", their implications are distinct, respectively pertaining to **supplementary knowledge** and **exemplar questions**.
>
> > Question B: In Figure 2, do numbers represent students' performance/accuracy in corresponding questions?
>
> Yes, they do. The numbers represent the accuracy in corresponding questions from 0 to 1.
>
> > Question C: Can you add an extended version of Table 4 that includes ±context, ordering effect, and topic subjects? I think it will be very interesting.
>
> Of course, this is indeed a good suggestion. Below is the extended table we have supplemented.
>
> **Without context:**
>
> | Models    | Disciplines       | SC           | SC(first)   | SC(last)    | MC           | TF           |
> |-----------|-------------------|--------------|-------------|-------------|--------------|--------------|
> | GPT-3.5   | STEM              | 0.393 (0.396)| 0.542 (0.570)| 0.403 (0.357)| 0.348 (0.454)| 0.546 (0.548)|
> |           | Social Science    | 0.440 (0.423)| 0.571 (0.584)| 0.407 (0.410)| 0.385 (0.496)| 0.517 (0.515)|
> |           | Humanity          | 0.551 (0.453)| 0.553 (0.563)| 0.357 (0.357)| 0.465 (0.549)| 0.505 (0.505)|
> |           | Others            | 0.617 (0.385)| 0.596 (0.607)| 0.296 (0.298)| 0.357 (0.429)| 0.560 (0.560)|
> | ChatGPT   | STEM              | 0.467 (0.468)| 0.395 (0.397)| 0.418 (0.395)| 0.272 (0.479)| 0.611 (0.566)|
> |           | Social Science    | 0.552 (0.537)| 0.520 (0.520)| 0.448 (0.445)| 0.317 (0.506)| 0.614 (0.604)|
> |           | Humanity          | 0.602 (0.503)| 0.506 (0.506)| 0.463 (0.447)| 0.303 (0.527)| 0.634 (0.624)|
> |           | Others            | 0.530 (0.519)| 0.500 (0.500)| 0.437 (0.424)| 0.312 (0.464)| 0.627 (0.604)|
> | GPT-4     | STEM              | 0.605 (0.595)| 0.691 (0.691)| 0.494 (0.473)| 0.546 (0.650)| 0.718 (0.716)|
> |           | Social Science    | 0.730 (0.724)| 0.788 (0.788)| 0.585 (0.627)| 0.607 (0.706)| 0.782 (0.770)|
> |           | Humanity          | 0.670 (0.668)| 0.743 (0.743)| 0.558 (0.565)| 0.524 (0.657)| 0.757 (0.752)|
> |           | Others            | 0.698 (0.698)| 0.839 (0.839)| 0.622 (0.587)| 0.446 (0.554)| 0.769 (0.769)|
>
> **With context**
>
> | Models   | Disciplines      | SC           | MC           | TF           |
> |----------|------------------|--------------|--------------|--------------|
> | GPT-3.5  | STEM             | 0.341 (0.448)| 0.298 (0.484)| 0.595 (0.595)|
> |          | Social Science   | 0.363 (0.536)| 0.246 (0.544)| 0.604 (0.604)|
> |          | Humanity         | 0.503 (0.595)| 0.357 (0.619)| 0.663 (0.663)|
> |          | Others           | 0.509 (0.469)| 0.223 (0.460)| 0.687 (0.687)|
> | ChatGPT  | STEM             | 0.593 (0.581)| 0.298 (0.499)| 0.675 (0.656)|
> |          | Social Science   | 0.623 (0.624)| 0.376 (0.582)| 0.737 (0.735)|
> |          | Humanity         | 0.647 (0.615)| 0.351 (0.589)| 0.748 (0.752)|
> |          | Others           | 0.596 (0.598)| 0.268 (0.451)| 0.754 (0.739)|
> | GPT-4    | STEM             | 0.594 (0.608)| 0.570 (0.679)| 0.738 (0.734)|
> |          | Social Science   | 0.671 (0.759)| 0.637 (0.733)| 0.807 (0.803)|
> |          | Humanity         | 0.531 (0.797)| 0.541 (0.686)| 0.797 (0.792)|
> |          | Others           | 0.626 (0.754)| 0.455 (0.580)| 0.784 (0.776)|
>
> Since we did not conduct experiments on option ordering under the context condition, the second table lacks the additional columns for the other two single-choice settings. Thank you for your insightful suggestions. From this informative extended table, we can derive the following conclusions, which are consistent with the conclusions in the paper:
>
> * Models perform better with the earlier appearance of the correct answers (Primacy Effect).
>
> * Models achieved better scores in social science and humanity. However, without human results as references, we can not ascertain whether such outcomes stem from the difficulty of STEM questions, as shown in Table 1 in the paper.
>
> * Models perform best on SC and worst on TF, because of more thinking steps.
>
> > Question D: Can you add your insights in the Discussion section on why the LLMs perform worse in intermediate-level Bloom's Taxonomy questions and how to improve the models' performance?
>
> Thank you for your valuable suggestions. Here, we have a few thoughts that you might find insightful.
>
> * Firstly, lower dimensions such as "remembering", "understanding" and "factual knowledge" tend to be more manageable for models. This is due to the auto-regressive training objectives of these models, which ensure strong memory capabilities.
>
> * Secondly, in the intermediate dimensions like "applying" and "procedural knowledge", application-based questions are included. These involve tasks like solving specific mathematical problems, making deductions using chemical theorems, engaging in logical reasoning, and more. Such questions are prone to errors and are inherently challenging for models.
>
> * Lastly, as for higher dimensions "evaluating" questions, the formidable linguistic capabilities of models indeed allow them to excel in these tasks, and perform even better than intermediate-level questions.
>
> These are some of our conjectures. The reason for such a distribution stems from the models' proficiency in natural language. Future models should enhance their capability for reasoning and problem-solving.
>
> > Adding horizontal lines in Table 4 will improve the readability and consistency with other tables.
>
> Thank you a lot for helping us improve our readability. We will proceed to revise the table accordingly.
>
> **[Final remark]** Once again, we extend our gratitude for your acknowledgment of the insights within our article and for the highly beneficial suggestions you provided. As many discussions and details are hidden due to the short paper's page limitation, we earnestly hope that our responses have addressed your concerns regarding the experiments, and we would appreciate it if you could consider raising the soundness score. Should you have further questions, we sincerely invite you to engage with us.
>
> Ref:
>
> [1] Brown, Tom, et al. "Language models are few-shot learners." Advances in neural information processing systems 33 (2020): 1877-1901.

---

### Meta-Review · Area_Chair_ZVkV · 2023-09-18

**Recommendation:** 4

**Metareview:**

This paper seeks to understand LLMs by applying a human test set based on Bloom's Taxonomy of Learning. Reviewers find that the approach has significant merit, but highlight weaknesses related to reporting information about the dataset, including annotator training and agreement, and robustness to instruction wording. These concerns appear to have been addressed in the discussion period.

---

### Decision · Program_Chairs · 2023-10-07

**Decision:**

Accept-Findings

**Comment:**

This paper seeks to understand LLMs by applying a human test set based on Bloom's Taxonomy of Learning. Reviewers find that the approach has significant merit, but highlight weaknesses related to reporting information about the dataset, including annotator training and agreement, and robustness to instruction wording. These concerns appear to have been addressed in the discussion period.